# Mechanisms that Incentivize Data Sharing in Federated Learning

## Abstract

Federated learning is typically considered a beneficial technology which allows multiple agents to collaborate with each other, improve the accuracy of their models, and solve problems which are otherwise too data-intensive / expensive to be solved individually. However, under the expectation that other agents will share their data, rational agents may be tempted to engage in detrimental behavior such as *free-riding* where they contribute no data but still enjoy an improved model. In this work, we propose a framework to analyze the behavior of such rational data generators. We first show how a naive scheme leads to catastrophic levels of free-riding where the benefits of data sharing are completely eroded. Then, using ideas from contract theory, we introduce *accuracy shaping* based mechanisms to maximize the amount of data generated by each agent. These provably prevent free-riding without needing any payment mechanism.

## 1 Introduction

Data is a *non-rivalrous good*—once produced, it can be repeatedly used multiple times without exhaustion. Thus, multiple firms can simultaneously use the data produced by any individual firm, increasing societal utility/welfare [21]. To promote such multiple usage, data portability requirements have been widely legislated, e.g., the GDPR in the EU, CCPA in California, etc [35]. As a consequence, services are required to enable a user to download any personal data collected and potentially re-upload it to a different service. These desiderata form a solid economic and legal basis for federated learning—a new paradigm in machine learning wherein multiple data-generating agents collaborate with each other to train a model on their *combined* data so that all the agents end up with a better model than they would have obtained on their own [22]. Such collaborative data sharing is already common in genomics research [51], internet advertisement targeting [18], and is also gaining traction between networks of hospitals [see, e.g., 45, 52, 41, 16].

It is clear that once a certain amount of data has been produced, privacy issues aside, societal welfare is maximized by allowing free access to the data thereby making it a public good. However, under such an expectation, a rational agent may be tempted to *free-ride*, i.e., consume the benefits of the data production by others without contributing any data themselves. This may lead to a collapse in the data generation with everyone wanting to free-ride. Such a problem inevitably arises with any public good [5]. Further, even if no agent actually free-rides and everyone intends to contribute data out of altruism, the mere perception that others may be free-riding reduces pro-social behavior and willingness to contribute [11]. Thus, the long-term success of federated learning in particular and data portability in general critically require overcoming free-riding. This motivates our main question:

> *How do we design a system which incentivizes rational agents to contribute their*
> *fair share of data, thereby maximizing the value of the resulting model and improv-*
> *ing collaboration?*

**Contribution and summary of results.**

- We formulate a principal-agent model [31] where each agent has a cost associated with generating a data point and wants to improve the value of a model while minimizing said costs (Sec. 2). Our formulation borrows ideas from contract theory while introducing new concepts that are specific to the federated learning setting.
- Using this framework we show how giving unconditional benefit of the combined data to all agents (as is standard in federated learning) leads to catastrophic free-riding where almost none of the agents contribute any data (Sec. 3) at their optimal responses.
- Accordingly, we propose to tune the value of the model received by an agent to their contribution. In the full-information setting when the agent's cost of data generation is known, we derive an optimal mechanism which overcomes free-riding and leads to maximal collaboration and data generation (Sec. 4).
- Finally, if the costs of an agent are unknown, we show (in App. D) how to design truth-revealing value curves at some cost to the principal (i.e., information rent) to incentivize the agents to report their true costs.

Our framework can capture free-riding and the need for collaboration when faced with challenging learning problems. The latter is novel to our framework—we show that if the learning task is too challenging, then it is not economically viable for any single agent to tackle the problem. However, using incentivizing data-sharing mechanisms, it may be possible to share the costs among participants and solve it collaboratively.

# 2 Modeling an Individual Agent

We begin with modelling the learning task and objective for an individual agent. We then provide a characterization of the optimal data contribution for each single agent without participating in a federated learning scheme.

## 2.1 Value of data

There are $n$ agents all of whom want to solve a *common* learning problem. This is often true in federated learning since coalitions form around solving some particular task. Concretely, we assume that all agents want to maximize a value function, $v(\mathcal{D}) : 2^{\mathcal{D}} \to [0, 1]$, for a dataset $\mathcal{D}$. For simplicity, we assume that every datapoint is *exchangeable* i.e. every datapoint has the same value as any other datapoint. While this is a strong assumption, it holds true if the data is generated by manually labelling a subset of an already public unlabelled dataset, as is common in machine learning; e.g., Cifar [30] and ImageNet [43]. This assumption is arguably also valid in our autonomous driving example where each data point involves a random path taken under random external conditions. With this, we can simplify the value function $v(\cdot)$ to depend only on the *size* of the dataset $m = |\mathcal{D}|$. For convenience, we will treat dataset sizes as a continuous real. Thus, every agent wants to maximize

$$v(m) : \mathbb{R}_{\geq 0} \to [0, 1] = \max(0, b(m)), \text{ where } b \text{ is continuous, non-decreasing and concave.} \quad (1)$$

We also assume w.l.o.g that $v(0) = 0$ and $\lim_{m \to \infty} v(m) > 0$. Perhaps the most natural way of defining the value of data is via the test accuracy obtained by training a model on the data. For example, each accurate product recommendation may lead to a sale or correct digital ad placement may lead to a click and hence ad revenue. This is also true if each error represents costly consequences. Each error by a medical diagnostic model, a loan application evaluation model, or an autonomous driving model may lead to significant suffering. In all of these cases, the value of data comes by directly improving generalization and guaranteeing test accuracy.

## 2.2 Agent's objective and optimal solution

Each agent $i$ has a marginal fixed cost $c_i > 0$ for producing a data point. Their cost for producing a dataset $\mathcal{D}$ with $m$ number of data points is then:

$$cost_i(m) = c_i m. \quad (2)$$

When manually labelling a dataset or when training an autonomous-driving model, this cost $c_i$ may represent the time spent by researchers/employees or an amount paid to crowd-sourced workers. The cost $c_i m$ may also represent the risk associated with privacy loss for the agent for revealing $m$ of their data points. By incurring this cost, they can obtain a model with value $v(m)$. Thus, the net utility of an agent is improve value for the least cost; i.e., to maximize

$$u_i(m) = v(m) - c_i m. \quad (3)$$

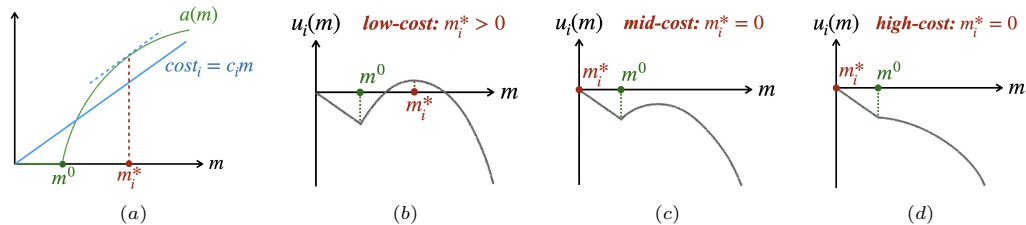

Figure 1: Illustration of the optimal amount of data for a single agent. *(a):* Value and cost versus the dataset size.*(b)-(d):* Utility function of a low/mid/high-cost agent versus the dataset size. Optimal amount for low-cost agent is positive but zero for mid and high cost agents.

**Theorem I** (Optimal individual generation)**.** *Consider an individual agent $i$ with marginal cost per data point $c_i$ and value function $v$ satisfying (1) working on their own. Then, the optimal amount of data $m_i^*$ is:*

$$m_i^* = \begin{cases} 0 & \text{if } \max_{m_i \geq 0} u_i(m_i) \leq 0; \\ \alpha_i^*, \text{ such that } b'(\alpha_i^*) = c_i & \text{otherwise.} \end{cases} \qquad (4)$$

*Further, for agents $i, j$ with costs $c_i \leq c_j$, their utility satisfies $u_i(m_i^*) \geq u_j(m_j^*)$ and $m_i^* \geq m_j^*$.*

As Figure 1 shows, if the learning problem is too hard ($m^0$ is large) or if the marginal cost $c_i$ is too high, the problem becomes infeasible for an individual agent to solve with $m_i^* = 0$. Such cases are especially important in federated learning where we want to enable agents to solve problems together which they cannot on their own. In other cases, the agent collects $m_i^* > 0$ data points.

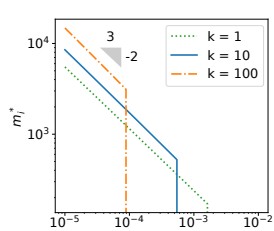

We can simulate the value function arising from the generalization guarantees of an ERM problem ([38, Theorem 11.8]) with $k$ measuring different difficulty levels (higher $k$ means that the learning problem is harder). Figure 2.2 shows the optimal data contribution $m_i^*$ versus the marginal cost $c_i$ for different number of total agents on a log-log scale in such a setting. We see that the optimal contribution decreases with cost as $m_i^* \propto c_i^{-2/3}$, matching the theory. The vertical

Figure 2: Optimal individual data contribution $m_i^*$ versus the marginal cost $c_i$ for different number of total agents.

lines indicate the cutoff for minimum viability—beyond this, the cost is too high for the problem to be solvable by an individual. This minimum viability cost is smaller for more harder problems (larger $k$), but the optimum contribution increases with increasing $k$ once this threshold is crossed.

## 3 Modeling Multiple Agents and Catastrophic Free-Riding

In this section we will study how agents behave when collaborating with each other as they do in federated learning. For this, we use a principal-*multi*-agent framework where the server who sets up the federated learning server is the principal.

### 3.1 Interaction between agents and server

The interaction between the federated learning server and the agents is formalized by a mechanism

$$\mathcal{M}(\mathbf{m}) : \mathbb{R}_{\geq 0}^n \to [0, 1]^n \text{ , which maps agents' contributions to values.} \qquad (5)$$

We assume that each agent $i$ generates and transmits $m_i$ data points to the server. Based on these contributions, the mechanism assigns models to the clients with differing valuations ; i.e., if agent $i$ contributes $m_i$ data points it receives a model with value $v_i \in [0, 1]$, where $\mathcal{M}(m_1, \ldots, m_n) = (v_1, \ldots, v_n)$. The interaction proceeds in three steps: (i) first the server publishes a mechanism $\mathcal{M}$, then (ii) each agent generates and transmits some data $m_i$ to the server, and finally (iii) the server returns a trained model to each agent following the mechanism. Note that the agents decide how much data to generate adaptively *after* knowing the mechanism $\mathcal{M}$. However, they do not have any bargaining power—they cannot re-negotiate the mechanism—but can only decide if they join or not. We also disallow monetary compensation or exchanges between the parties since implementing them adds additional complexity. The only guarantee is that the server truthfully executes the protocol $\mathcal{M}$.

Given that the server necessarily needs to follow through on the mechanism, we need to make sure the mechanism is implementable.

**Definition A** (**Feasible mechanism**). *A mechanism which returns value $[\mathcal{M}(\mathbf{m})]_i$ to agent $i$ is said to be* feasible *if for any $i \in [n]$ and any $\mathbf{m} \in \mathbb{R}^n_{\geq 0}$, it satisfies $[\mathcal{M}(\mathbf{m})]_i \leq v(\sum_j m_j)$.*

This is because we can pool together all the agent contributions $\mathbf{m}$ and train a model to value $v(\sum_j m_j)$. Since $v(\cdot)$ is monotone, this is an upper bound on the value which can be obtained. However, it is always possible to use a subset of this data, or degrade the model in a controlled way using noisy perturbations. Thus, this captures mechanisms which are implementable in practice.

Faced with a potential feasible mechanism $\mathcal{M}$, an agent has to decide whether to join or simply train on their own. A mechanism which offers an especially bad value would discourage an agent and they would likely leave the server and train on their own. We will formalize this next.

**Definition B** (**Individual rationality (IR)**). *Given data contributions $\mathbf{m}$ by the $n$ agents with costs $\mathbf{c}$, the mechanism provides a model with value $[\mathcal{M}(\mathbf{m})]_i$ to agent $i$. Such a mechanism $\mathcal{M}$ is said to satisfy IR if for any agent $i \in [n]$ and any contribution $\mathbf{m}$,*

$$[\mathcal{M}(\mathbf{m})]_i - c_i m_i \geq v(m_i) - c_i m_i \,. \tag{6}$$

A mechanism which satisfies individual rationality guarantees that for any agent the value of the model received (and hence their utility) will be no worse than if they trained on their own. Since IR guarantees that all rational agents will participate in our mechanism, and participation is key to success of any platform, we will restrict our focus henceforth to mechanisms which satisfy IR.

Given any mechanism $\mathcal{M}$, we would like to argue about how rational agents would respond and how much data they would contribute. For this, we use the notion of an equilibrium.

**Theorem II** (Existence of pure equilibrium). *Consider a feasible mechanism $\mathcal{M}$ which can be expressed as:*

$$[\mathcal{M}(m_i; \mathbf{m}_{-i})]_i = \max(0, \nu_i(m_i; \mathbf{m}_{-i})) \,,$$

*for a function $\nu_i(m_i; \mathbf{m}_{-i})$ which is* continuous *in $\mathbf{m}$ and* concave *in $m_i$. For any such $\mathcal{M}$, there exists a pure Nash equilibrium in data contributions $\mathbf{m}^{eq}(\mathcal{M})$ which for any agent $i$ satisfies,*

$$[\mathcal{M}(\mathbf{m}^{eq}(\mathcal{M}))]_i - c_i m_i^{\mathcal{M}} \geq [\mathcal{M}(m_i, \mathbf{m}^{eq}(\mathcal{M})_{-i})]_i - c_i m_i \,, \text{ for all } m_i \geq 0 \,. \tag{7}$$

Thus, under reasonable conditions on the mechanism $\mathcal{M}$ which are satisfied for all the mechanisms we consider, an equilibrium always exists such that no agent can improve their utility by unilaterally changing their contribution. If all players are rational, then such an equilibrium point is a natural attractor with all the agents gravitating towards such contributions. Thus, it is reasonable to use the data contributions at this equilibrium to evaluate and compare different mechanisms.

Note that the mechanism is not concave because of the presence of a $\max(0, \cdot)$, and the resulting utilities of the agents are not even quasi-concave. Despite this, our proof uses the specific properties of our setting to prove existence. Our techniques may be more broadly applicable to study non-concavities arising from "minimum viability".

## 3.2   Free-riding in the standard federated setting

We now examine the behavior of rational agents in the standard federated learning. Returning a model trained on the combined dataset to everyone corresponds to the mechanism

$$\mathcal{M}(\mathbf{m}) = \left(v(\textstyle\sum_j m_j), \forall i \in [n]\right) \,. \tag{8}$$

Clearly, this mechanism is feasible (Def. A) and also satisfies individual rationality (Def. B) since the value function $v(\cdot)$ is non-decreasing and $\sum_j m_j \geq m_i$ for any $i \in [n]$. In fact, given a data contribution $\mathbf{m}$, this mechanism may maximize the utility for all agents. This observation may at first seem like a strong argument in favor of this standard scheme. However, recall that the agents choose their contribution $\mathbf{m}$ *after* the server publishes the mechanism $\mathcal{M}$. Thus, we need to first analyze how much data rational agents would contribute.

**Theorem III** (Catastrophic free-riding). *Consider $n$ agents with costs $\{c_i\}$ with a unique least cost agent $c_{\min} = \min_i c_i$. Let $\{m_i^*\}$ be the equilibrium contributions of agents when alone. The standard federated learning mechanism corresponding to $[\mathcal{M}(\mathbf{m})]_i = v(\sum_j m_j)$ for all clients $i$ is feasible and IR, and has an* unique *equilibrium. At this equilibrium, only the lowest cost agent contributes:*

$$m_i^{eq} = \begin{cases} m_i^* & \text{if } c_i = c_{\min} \\ 0 & \text{otherwise.} \end{cases} \tag{9}$$

The agent with the least cost $c_{\min} = \min_i c_i$ would have collected $m^*_{\min}$ amount of data on their own. For any other agent $i$, $c_i \geq c_{\min}$ and so $m^*_i \leq m^*_{\min}$. Thus, agent $i$ would already have access to data sufficient to satisfy them by the federated learning mechanism. The increase in value $v(\cdot)$ for collecting an additional data point beyond this is less than the marginal cost $c_i$ incurred. This results in catastrophic free-riding, with only a single agent collecting data.

**Remark 1** (Collapse of collaboration). *Consider the case where $m^*_i = 0$ for all agents $i$, either because the learning problem is too hard or because the cost of data collection is too high for any individual agent. Theorem III implies that* no data *will be collected even with collaboration. Thus, if a problem is too costly to solve by an individual, it will remain insurmountable via standard federated learning. This is because everyone rationally assumes that everyone else will free-ride, defeating the main motivations of federated learning.*

# 4 Value Shaping under Verifiable Costs

How do we design mechanisms which prevent free-riding? In this section we will study this question assuming everyone (the server and the agents) know the costs **c** involved in producing the data (we study the unknown costs setting in Section D), or that the costs can be verified; i.e., the agent cannot incur cost $c$ and report a different cost $\tilde{c}$. This is justifiable in some cases—the cost of labelling a data point by a crowd-worker can be estimated by all parties. We formalize our goal of data maximization and give a simple optimal mechanism for it.

## 4.1 Value shaping mechanism

A mechanism $\mathcal{M}$ is data-maximizing given costs **c** if it maximizes the data collected at equilibrium.

**Definition C** (**Data Maximization**). *Suppose that given a mechanism $\mathcal{M}$, let $\mathbf{m}^{eq}(\mathcal{M})$ correspond to the amount of data generated by the agents at equilibrium. $\hat{\mathcal{M}}$ is* data-maximizing *if it maximizes the amount of data collected at equilibrium*

$$\hat{\mathcal{M}} \in \arg\max_{\mathcal{M}} \sum_j [m^{eq}(\mathcal{M})]_j, \qquad (10)$$

*subject to $\mathcal{M}$ being feasible and satisfying IR.*

Figure 3: Illustration of value shaping. *(red curve)*: model value returned to agent $i$ by the mechanism; *(grey curve)*: model value for agent $i$ without participation; *(green curve)*: model value if agent $i$ receives all the data from the other agents.

**Mechanism description.** If we give $\Delta m_i$ free data to agent $i$, then at equilibrium they will reduce the data they generate—they will only generate $(m^*_i - \Delta m_i)$ additional data. To prevent this, our key insight is to condition the amount of extra data on their actual contribution. For a given set of costs **c** and some small $\varepsilon > 0$, consider the following mechanism:

$$[\mathcal{M}(\mathbf{m})]_i = \begin{cases} v(m_i) & \text{for } m_i \leq m^*_i \\ v(m^*_i) + (c_i + \varepsilon)(m_i - m^*_i) & \text{for } m_i \in [m^*_i, m^{\max}_i] \\ v(\sum_j m_j) & \text{for } m_i \geq m^{\max}_i, \end{cases} \qquad (11)$$

where $m^{\max}_i$ is defined such that $v(m^{\max}_i + \sum_{j \neq i} m_j) = v(m^*_i) + (c_i + \varepsilon)(m^{\max}_i - m^*_i)$. We illustrate the mechanism in Figure 3. Even without any external incentivization, agent $i$ will compute $m^*_i$ data points. Thus, for $m_i \leq m^*_i$ (10) returns a model trained on solely their own data. After $m^*_i$, however, the marginal gain in value becomes smaller than the additional cost $c_i$. Hence, the agent requires active incentivization here and (10) ensures that for every additional data point computed, the marginal gain in value is strictly more than the cost $c_i$. However, the mechanism cannot provide unlimited value either and has to remain feasible, giving us our final constraint.

## 4.2 Analysis

**Theorem IV** (Data maximization with known costs). *The mechanism $\mathcal{M}$ defined by (11) is data-maximizing for $\varepsilon \to 0^+$. At equilibrium, a rational agent $i$ will contribute $m^{\max}_i$ data points where $m^{\max}_i \geq m^*_i$, yielding a total of $\sum_j m^{\max}_j$ data points.*

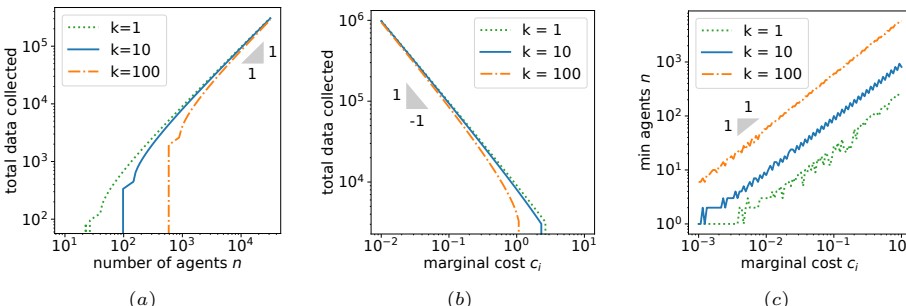

(a)                                      (b)                                      (c)

Figure 4: Equilibrium simulations of our mechanism: the total data collected at equilibrium (a) increases linearly with the number of agents $n$, (b) decreases as $c_i^{-1}$ with increasing marginal cost per data point, and is relatively unaffected by the complexity $k$. (c) The number of agents required to cross the minimum viability threshold (i.e. smallest $n$ for which $m_i^{\max} > 0$) increases linearly with both the marginal cost $c_i$ and complexity $k$. Optimal individual contribution for all settings is $m_i^* = 0$; i.e., no data would be generated by standard federated learning.

Thus, to encourage collaboration we can set up a central repository of the data to which each agent is required to contribute. The cost for collecting a data point (say $c$) can easily be estimated and assumed to be the same for all agents. Using this estimate, we can compute a threshold. The agents don't receive any additional data for contributing up to this threshold. For each data point contributed beyond the threshold, the agents receive an increasing amount of additional data. By Theorem IV, this would prevent free-riding by the agents and ensure the best trained model reaches the consumers.

**Remark 2** (Deterrence). *At equilibrium, mechanism $\mathcal{M}$ in (11) ensures all agents contribute $m_i^{\max} \geq m_i^*$; i.e., they generate* more *data than they would on their own. Further, every agent receives a model trained on this combined dataset with value $v(\sum_j m_j^{\max})$. One can view our mechanism as using a deterrent which punishes free-riding, ensuring that all agents fully utilize the combined data. At equilibrium, such a deterrent is never actually invoked but just forms a credible threat.*

Suppose all agents have the same cost $c$. Theorem IV shows that the mechanism collects $nm^{\max}$ data points in total. However, $m^{\max}$ also depends on $n$. This is because with a larger pool of data contributions, the server can more strongly incentivize an individual and extract more data. There is a natural ceiling to this though—the value caps at 1. Thus, the absolute maximum data that can be extracted from an individual agent is $m$ which satisfies $v(m^*) + c(m - m^*) = 1$. This gives us the range for the total data contributions to be $[nm^*, n(m^* + {}^{(1-v(m^*))}/c)]$.

**Remark 3** (Collaboratively overcoming minimum viability). *When $m^* = 0$, i.e., the problem is not solvable by an individual agent, the net contribution from our mechanism $nm^{\max}$ may still be positive. Suppose that the cost for all agents is the same $c$. Then, the total data collected is $m^{tot}$ which satisfies*

$$^{c}/_{n} \cdot m^{tot} = v(m^{tot}) .$$

*This implies that for sufficiently large $n$, the cost $c$ is successfully shared and we obtain a positive data contribution. However, note that $m^{tot} = 0$ is also a valid solution and remains an equilibrium. If all other agents don't contribute, there is no extra data to share and so there is no incentive to compute extra data. In practice, this undesirable equilibrium is unlikely to be encountered since it has lower utility. It can also be prevented by the platform itself taking part as an agent and committing to non-zero data collection.*

Empirically, in Figure 4 we compute the equilibrium for value shaping assuming the value of the data is test accuracy. We assume all agents have the same cost, and observe the effect on the equilibrium data contribution as we vary the cost $c$ and the total number of agents $n$. We used the following default parameters unless otherwise states: optimal value of $a_{opt} = 0.95$, marginal cost $c_i = 0.1$, participants $n = 10^4$. Under all parameter configurations of this experiment, the optimal individual contribution is $m_i^* = 0$, while the equilibrium data contributions are significantly larger as expected, validating our theory.

Finally, in App. B we discuss the incentive compatibility of our schemes, App. C performs additional simulations and detailed comparisons with prior work, and in Appendix D we extend our framework to the setting of unverifiable costs. Our conclusions translate to this unverifiable cost setting as well.

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

# Appendix

## Contents

# A  Review on the Related Work and Contract Theory Background

The literature on mechanism design and federated learning is vast. We discussed the most closely related work in three verticals in the main text; we include a detailed review of the broader literature in this section.

Over the past decade, federated learning (FL) has emerged as an important paradigm in modern large-scale machine learning [28, 27, 22, 32, 41]. Specifically, FL research has resulted in many applications to overcome practical challenges such as data silos and data sensitivity: on one side, since more training data often gives better model performance, data silos results in scarcity of labeled training data and puts limit on the industrial performance; on the other side, in high-stakes applications the data may contain private user information and thus the sharing of data is constrained by regulations and laws [49, 4, 10, 35]. Given these challenges, FL provides a useful scheme for different agents / parties to train collaboratively and leverage the benefit from other agents' data, while the training data remains distributed over the agents. Such a framework has been shown to be able to bring improved model performance to all the participants. Indeed, many prior works have been devoted to develop more scalable and communication-efficient distributed optimization algorithms for FL [29, 37, 8].

However, one cannot ignore an important aspect in the standard FL scheme, which is the incentives aspect. The standard FL scheme may incentivize strategic agents to contribute less data in order to minimize their data collection cost and maximize the gain from participating in the federated learning mechanism. Although the participation and contribution of each agent is often legislated by certain protocols, such free-riding behavior has been notoriously hard to regulate and prevent in practice [17, 19]. Recently, a few works have started to explore such free-riding behavior in FL, with various incentive models proposed [40, 44, 34, 17, 13, 55]. However, the majority this work has focused on a taxonomy of free-rider attacks or the detection of attacks under the existing FL scheme, instead of proposing mechanisms that incentivize maximal data contribution. In this work, we strive for a mechanism for information sharing under the standard federated learning setting such that rational agents are incentivized to contribute their maximal amount of data.

In this work, we focus on the free-riding behavior of FL agents in terms of data collection. In FL, the data collection happen on the agents' side before they join the mechanism for training models. Therefore, the cost of collecting data is often *private* information to each agent. Such an information asymmetry brings difficulty to prevent free-riding, because the agents might simply report fake costs. This brings the need to design *incentive* mechanisms for FL, under which the agents are incentivized to behave truthfully, which is also guaranteed to lead to the best utility.

Indeed, designing incentive mechanisms under private costs is not new, and has been a main focus of the contract theory literature Smith [47], Laffont and Martimort [31], Bolton and Dewatripont [7]. Moreover, the existence of a central server (a "principal") in FL brings further convenience to apply a principle-agent model. An emerging line of recent works have been exploring the application of contract theory for federated learning [24, 25, 23, 33, 48, 12, 54]. In particular, Tian et al. [48] proposed a contract-based aggregator under a multi-dimensional contract model over two possible types of agents and showed improved model generalization accuracy under that contract. However, their mechanism focused on eliciting the private type information instead of maximizing the data contribution. To the best of our knowledge, our work is a first step to use contract theory for *data maximization* in federated learning. Further, prior work has focused on how to design payments to agents, rather than the value-shaping problem that we focus on here.

This work is related to the active line of research on mechanism design for collaborative machine learning, which involves multiple parties each with their own data, jointly training a model or making predictions in a common learning task [46, 53]. In collaborative machine learning, a major focus has been the design of model rewards (i.e., data valuation) in order to ensure certain fairness or accuracy objectives. Towards that goal, there has been model rewards proposed based on notions from the cooperative game theory literature such as the Shapley value [20, 50]. However, the guarantees of these model rewards depend on the assumption that the agents are already willing to contribute the data they have. In this work, we study a different incentivization task for data maximization.

More broadly, apart from data maximization, there are other objectives which are of interest in federated learning, such as fairness and welfare objectives, that have been under active study [15, 14, 39]. We defer a thorough analysis of the tradeoffs among various objectives to future research.

## B   Incentive compatibility under Verifiable Costs

One of our motivating reasons for preventing free-riding was to ensure that none of the participating agents feel taken advantage of. That is, we wish to satisfy some notion of fairness. However, there may potentially be new sources of unfairness in (11). In particular, consider two agents, $i, j \in [n]$, with different costs: if $c_i \leq c_j$ then $m_i^* \geq m_j^*$. Here, an agent $i$ with smaller cost $c_i$ faces two disadvantages under mechanism (11): (i) they have a larger threshold amount of data $m_i^*$ they have to contribute before receiving any benefit, and (ii) they receive a smaller increase in value $(c_i + \varepsilon)$ for each additional data point computed.

If the cost for generating each data point is inherently fixed (such as the cost of driving a vehicle) this is arguably not an issue. However, in many other settings an agent may innovate and develop new methods to reduce their cost of collecting a data point. In fact, the business model of large internet advertising providers is based on systems which can cheaply capture consumer data in order to show them better advertisements. Would our data-sharing mechanism (10) disincentivize agents from such innovations? We show this is in fact not true.

**Theorem V** (Incentive compatibility). *Under given costs* **c***, consider our optimal mechanism* (11) *with equilibrium contributions* $\mathbf{m}^{\max}$*, and agents working individually with equilibrium contributions of* $\mathbf{m}^*$*. The utility of the every agent* $i$ *remains unchanged:*

$$v(\textstyle\sum_j m_j^{\max}) - c_i m_i^{\max} = v(m_i^*) - c_i m_i^* \,.$$

Thus, our mechanism does not induce any distortions in the incentive structure. Further, recall by Theorem I, the utility $u_i(m_i^*) \geq u_j(m_j^*)$ if $c_i \leq c_j$. This implies that users with smaller costs continue to receive a higher utility, encouraging them to innovate and reduce the costs; i.e., our mechanism is incentive compatible. Of course this is assuming that the costs incurred by an agent is verifiable. They cannot lie about the true cost, but may be able to choose between different collection strategies.

**Remark 4** (Distribution of surplus). *One may ask where the additional surplus which is generated by agents collaborating has disappeared, since the agents receive none of it. Our mechanism utilizes this surplus in order to extract additional data,* $m_i^{\max} - m_i^*$*, from the agents. Thus, all the additional surplus goes into improving the value of the model and hence to the end consumers of the model.*

## C   Additional Simulations and Comparisons

### C.1   Simulating collaborative training of GPT-3

We illustrate through a pedagogical example how one may use our theory in practice. We first extract the data of loss values obtained by training GPT-like language models on dataset of varying sizes from [26, Fig. 1] using WebPlotDigitizer [42]. Then, we fit a simple linear regression model in the log-log space to obtain a close fit $\text{loss}(m) = (\frac{5.4 \times 10^{13}}{m})^{0.95}$. We use this to model how the loss would decrease as data increases. Then, we can define accuracy as $(1 - \text{loss}(m))$. To define the value function, we need to assign a dollar value to a perfectly trained model. Microsoft reportedly paid 1 billion ($10^9$) 2019 dollars for a license to GPT-3 [2] and hence this forms an estimate of the value of a fully trained model to one company. Thus we have

$$v(m) = 10^9 (1 - (\tfrac{5.4 \times 10^{13}}{m})^{0.95}) \,.$$

GPT-3 was trained on 500 billion tokens [9]. OpenAI has raised an estimated 1 billion USD [1]. Suppose that accounting for salaries of all personnel involved etc., we allocate half of this money as being spent on training GPT-3, this gives an estimate of the marginal cost per datapoint as $c_i = 10^{-3}$ USD. With these numbers, we see that we need at least 1000 companies (each for whom the trained model is worth 1 billion USD) collaborating together to make it feasible. Note that with our estimated costs and benefits, it seems like OpenAI is at a loss. This is true–it would likely need to license GPT-3 (or its sucessors) to many more companies before it breaks even. Finally, we emphasize that this was more of a pedagogical exercise and not an actual prediction about outcomes. The stylized framework here is meant to provide qualitative insights about the incentives at play.

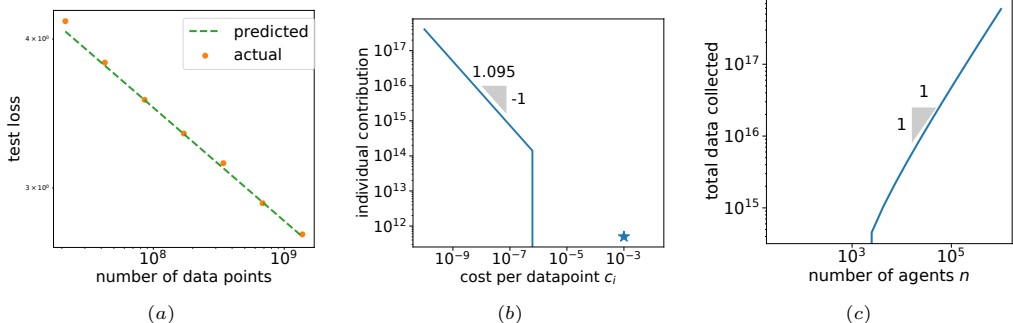

(a)  (b)  (c)

Figure 5: Simulating training of GPT-3. In (a), we take real world data (yellow dots) of how loss scales with data size and show that $(\frac{5.4\times10^{13}}{m})^{0.95}$ is a good fit (dashed green). We combine this with an estimate of the value of model and compute the optimal individual data contribution in (b) for different marginal data costs. The blue star shows the estimated marginal cost and total data collected by open-AI for training GPT-3. Finally, (c) shows the data collected by our data-maximizing mechanism with the estimated utility function. It shows that we need at least 1000 collaborative agents to train the GPT-3 model. The data collected initially grows super-linearly in $n$, but asymptotically becomes linear.

## C.2  Comparison with baselines

We compare with some alternative fairness inspired deterrence mechanisms which do not rely on our contract theory framework. These can be summarized as if you do not contribute as much as your peers, then you may be punished. Suppose the agents submit $\mathbf{m}$ number of data points. Then, the server chooses a feasible mechanism which returns a model of value less than $v(\sum_j m_j)$ to the client $i$. We consider the following mechanisms:

- **Proportionate data (PD).** Penalize agent $i$ if they submit less number of data points than their peers as

$$[\mathcal{M}(\mathbf{m})]_i = \left(\frac{m_i}{\max_j m_j}\right)^p v(\textstyle\sum_j m_j) \quad \text{for } p \in [0, 1] \tag{12}$$

- **Proportionate value (PV).** Penalize agent $i$ for contributing less value than their peers:

$$[\mathcal{M}(\mathbf{m})]_i = \left(\frac{v(m_i)}{\max_j v(m_j)}\right)^p v(\textstyle\sum_j m_j) \quad \text{for } p \in [0, 1] \tag{13}$$

- **Proportionate Shapley (PS) [46].** Shapely values has a long history of being used a fair contribution measure. Thus, we can compute the Shapely value for each agent's contribution $\phi_i(\mathbf{m})$ and penalize as

$$[\mathcal{M}(\mathbf{m})]_i = \left(\frac{\phi_i(\mathbf{m})}{\max_j \phi_j(\mathbf{m})}\right)^p v(\textstyle\sum_j m_j) \quad \text{for } p \in [0, 1]. \tag{14}$$

  If all other agents are contributing $m$ datapoints, the shapely value for agent $i$ for contributing $m_i$ datapoints simplifies as

$$\phi_i(m_i) = \frac{1}{n} \sum_{k \in [n]} v(km + m_i) - v(km).$$

In all cases, $p = 0$ returns to the standard federated learning scheme which, as we saw in Section 3, has catastrophic free-riding. These measures have numerous shortcomings which we explore in sequence.

### C.2.1  Many bad equilibria

Because the mechanism only penalizes on relative performance among the different agents, there are multiple stable equilibrium. Consider $n$ identical agents with same cost $c$ here in a *low-cost* setting with positive optimal individual contributions $m^* > 0$. Our conclusions also hold in more general settings, but we focus on this setting for simplicity.

**Theorem VI.** *Consider mechanisms* (12)–(14) *with $n$ identical agents with marginal cost $c$. Let $m^* > 0$ be the equilibrium individual contribution and $m^{\max}$ is the equilibrium contribution by our optimal mechanism. Then, there is a set of data contributions $\mathcal{S}$ such that all agents contributing $m \in \mathcal{S}$ constitutes an equilibrium. Further, $\frac{m^*}{n} \in \mathcal{S}$ is an equilibrium with the maximum utility.*

Note that this implies that every agent only contributing $\frac{m^*}{n}$ i.e. $n$ times lesser than they would on their own is also an equilibrium. With this only $m^*$ datapoints would be collected by the server. Further, this equilibrium corresponds to the maximum utility and so it is possible that all agents will converge to this. In contrast, our optimal scheme has an unique equilibrium corresponding to maximum data contribution.

*Proof.* Consider the generic mechanism $[\mathcal{M}(\mathbf{m})]_i = \left(\frac{\psi_i(\mathbf{m})}{\max_j \psi_j(\mathbf{m})}\right)^p v(\sum_j m_j)$ for any positive, continuous, non-decreasing contribution measure $\psi$. Suppose that all agents submit $m$ data-points. This is an equilibrium if the following condition is satisfied:

$$-p\frac{\psi_i'(m)}{\psi_i(m)}v(nm) \geq v'(nm) - c \geq 0\,.$$

The right hand side is satisfied for $m = m^*/n$. Also note that $\psi'$ is positive meaning the left hand hand side is negative. This implies that as long as $\psi'$ and $psi$ are continuous around $m$, there exist a set of solutions all of which satisfy the above condition. Contributing $m = m^*/n$ is utility maximizing for all the agents in general, and so corresponds to the maximum utility equilibrium as well.  $\square$

One way out of this may be for the server to take part in the process as an agent and also contribute data. This way by increasing its contribution, the server can force other agents choose an equilibrium corresponding to a large equilibrium.

### C.2.2   Sensitivity to choice of $p$ and under-performance

Consider $n$ agents each have identical *high-costs* $c$ with optimal individual contribution is $m^* = 0$. We use the same experimental setup as in Figure 4 with $a_{opt} = 0.95$, marginal cost $c = 0.1$, and $n = 10^4$. In Figure 6, we numerically compute the equilibrium corresponding to the *maximum data contribution* for each of alternative mechanisms, assuming the server may be able to intervene and direct the agents towards the most beneficial equilibrium.

When we use $p = 0$ all the alternative schemes recover the standard FedAvg scheme. As we saw in Sec. 3, this implies for $p = 0$ there is catastrophic free-riding and hence the equilibrium contribution is 0. However, if we choose a value of $p$ too large, it is possible that the mechanism no longer satisfies *individual rationality*. This means that at equilibrium the agents drop out and effectively contribute 0 data points again. The proportional value scheme significantly under performs, whereas the proportional Shapley scheme suggested by Sim et al. [46] and the much simpler proportional data scheme both perform reasonably well. However, note that even for the best value of $p$, both these schemes do not match our much simpler data-maximizing value shaping mechanism.

### C.2.3   Discrimination against high-cost agents

Consider $n = 10^4$ agents with $a_{opt} = 0.95$ and which are one of two types: either they have a low cost of $1 \times 10^{-4}$, or they have a comparable but slightly higher cost of $2 \times 10^{-4}$. A fraction (say $p_i \in [0,1]$) are of the low cost and the rest have high cost. We assume the server is aware of (or can verify) the cost of each of the agent. In Figure 7, we numerically compute the equilibrium data contribution of the high and low cost agents for the data maximizing value shaping mechanism, and the proportional data mechanism. For the latter PD, we compute the *maximum data contribution equilibrium* as well as use the *optimal $p$* to compare value shaping against the best possible version of the alternative mechanism.

We observe that the total amount of data collected by value shaping is much more (up to $50\times$) than PD, especially when a large fraction of agents have high cost. The high cost agents continue to contribute significant amount of data when using the value shaping mechanism. Recall that with these contribution levels, they receive full value of the combined data from the mechanism. However, very starkly, the high cost agent chooses to opt-out and contributed no data with the PD mechanism. This means that with PD, the high cost agents *receive zero value*. This is a direct result of PD being

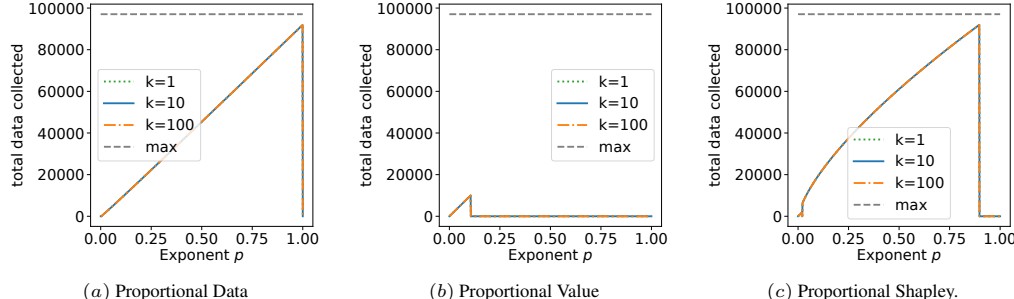

(a) Proportional Data       (b) Proportional Value       (c) Proportional Shapley.

Figure 6: Total data collected as we vary the exponent $p$ in the different mechanisms. The black dashed line represents the data collected by our data maximizing value-shaping mechanism outlined in Sec. 4. Using a small $p$ is an insufficient deterrent and so each agent tends to free-ride yielding low overall data collection. Using too high value of $p$ (say $p = 1$) makes the deterrent so strong that the agents rationally chooses to drop out and contribute 0.

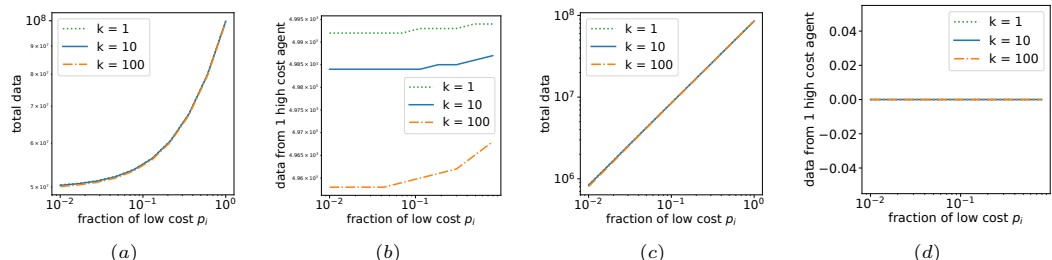

(a)       (b)       (c)       (d)

Figure 7: Data contributions in a mixture of low and high cost agents as we vary the fraction of low cost agents. Left shows the data maximizing value shaping mechanism and the (a)total data collected, and (b) data contributed by the high cost agents. Right (c) and (d) show the same with proportional data mechanism. Total data collected in (a) with value shaping is $50\times$ more than in (c) with proportional data. Further, the high cost agents continue to contribute and receive full value in (b) with value shaping, but they opt-out and receive zero value in (d) with proportional data. Thus, not accounting for costs may end up discriminating against high cost agents.

agnostic to the cost–it penalizes an agent for contributing lower amount of data without taking into consideration the difference in costs involved in such a process. In the face of such deterrent, the high cost agent rationally chooses to out-out. This is despite the costs differing by a mere $2\times$.

Our framework has thus thrown to light an important insight–'fair' mechanisms which do not account for difference in costs of the agents will end up being extremely unfair. Further, the high cost agents will rationally chose to drop out and derive no value from the system. Consider the setting where agents from resource-rich and resource-poor locations collaborate. The latter agents will have a higher data collection costs due to systemic barriers and will be unfairly penalized in a proportionate fairness approach.

## D  Data Maximization under Unverifiable Costs

Until now, we assumed that the cost of all agents is known to everyone involved, or is atleast verifiable. In some settings where the costs are universal and outside the control of the agent, this assumption may be justified. However, in numerous other cases, the exact process of the data generation may be a trade secret and so there is uncertainty about the cost incurred by an agent. In this section, we examine how to incorporate such uncertainties into our mechanism.

We focus on the simplest version of this uncertainty. Suppose that we know that the cost of each agent can either be low ($\underline{c}$) or high ($\bar{c}$). Further, suppose we have some prior knowledge where agent $i$ has low cost $\underline{c}$ with probability $p_i$ and $\bar{c}$ with $(1 - p_i)$. Note that there is an inherent *information asymmetry* in this setting. The agent knows the realization of their cost, $c_i \in \{\bar{c}, \underline{c}\}$, whereas the server only knows the distribution from which it was drawn. In particular, the server needs to present a mechanism $\mathcal{M}$ to an agent without knowing their actual cost.

### D.1  Mechanism description

Suppose each agent independently selects their cost to be low ($c_i = \underline{c}$) with probability $p_i$. Let an agent with low cost $\underline{c}$ generate $\underline{m}^*$ data points at equilibrium on their own (and correspondingly define $\bar{m}^*$ for a high-cost agent). Then, for some small $\varepsilon > 0$ and $\bar{m}^* \leq m_i^\uparrow \leq m_i^\downarrow$, consider the following mechanism (illustrated in Figure 8)

$$[\mathcal{M}(\mathbf{m})]_i = \begin{cases} v(m_i) & \text{for } m_i \leq \bar{m}^* \\ v(\bar{m}^*) + (\bar{c} + \varepsilon)(m_i - \bar{m}^*) & \text{for } m_i \in [\bar{m}^*, m_i^\uparrow] \\ v(m_i^\downarrow + \sum_{j \neq i} m_j) - (\underline{c} + \varepsilon)(m_i^\downarrow - m_i) & \text{for } m_i \in [m_i^\uparrow, m_i^\downarrow] \\ v(\sum_j m_j) & \text{for } m_i \geq m_i^\downarrow . \end{cases} \quad (15)$$

Recall from Theorem I that $\underline{m}^* \geq \bar{m}^*$ since $\underline{c} \leq \bar{c}$. Thus, agents with either costs do not need additional incentive to collect data up to $\bar{m}^*$. Now, consider a high-cost agent. After $\bar{m}^*$, they need a marginal gain in value of at least $\bar{c}$ which they do not get on their own. Additional supplementary data is provided by (15) until $m_i^\uparrow$ to incentivize a high-cost agent. It is now in their best interest to contribute $m_i^\uparrow$. For the low-cost agent, the marginal gain in value is at least $\underline{c}$ until $m_i^\downarrow$, making this their best contribution. The specific values of $m_i^\downarrow$ and $m_i^\downarrow$ (points D and C) can then be chosen to maximize the expected data contribution $((1 - p_i)m_i^\uparrow + p_i m_i^\downarrow)$.

For $\Delta m_{-i} := \sum_{j \neq i} m_j$, let $\bar{m}^{\max}$ be the maximum amount of data a high-cost agent can be incentivized to contribute as in (11) i.e. it is defined to be $v(\bar{m}^{\max} + \Delta m_{-i}) = v(\bar{m}^*) + \bar{c}(\bar{m}^{\max} - \bar{m}^*)$, and $\underline{m}^{\max}$ defined correspondingly for the low-cost agent. Then, we define $m_i^\downarrow$ (point D) to satisfy

$slope_{AC} = \bar{c} + \epsilon$
$slope_{CD} = \underline{c} + \epsilon$
– acc. with all data
– acc. no sharing

Figure 8: value shaping mechanism under unknown costs. *(red curve)*: model value returned to agent $i$ by the mechanism; *(grey curve)*: model value for agent $i$ without participation; *(green curve)*: model value if agent $i$ receives all the data from the other agents.

$$v'(m_i^\downarrow + \Delta m_{-i}) = \min\left(\max\left(\underline{c} - \tfrac{p}{1-p}\bar{c}, \; v'(\underline{m}^{\max} + \Delta m_{-i}), \; v'(\bar{m}^{\max} + \Delta m_{-i})\right)\right). \quad (16)$$

Then, we can define $m_i^\uparrow$ (point C) as the intersection of the two linear curves (starting from A and D in Fig 8):

$$v(m_i^\downarrow + \sum_{j \neq i} m_j) - (\underline{c} + \varepsilon)(m_i^\downarrow - m_i^\uparrow) = v(\bar{m}^*) + (\bar{c} + \varepsilon)(m_i^\uparrow - \bar{m}^*). \quad (17)$$

Note that our mechanism withholds some data from a high-cost agent resulting in a lower value model for them. This is necessary to prevent a contribution level targeted at high-cost agent from becoming attractive to a low-cost agent.

## D.2 Analysis

We now analyze the properties of our expected data-maximization algorithm.

**Theorem VII** (Expected data maximization). *Mechanism* (15) *is feasible, satisfies IR, and has a unique Nash equilibrium:* $m_i^{eq} = m_i^{\uparrow}$ *if* $c_i = \bar{c}$ *and otherwise* $m_i^{eq} = m_i^{\downarrow}$. *Further, for* $\varepsilon \to 0^+$, *the mechanism* (15) *maximizes the expected (over the sampling of the true costs) amount of data collected with*

$$\sum_j (1 - p_j) m_j^{\uparrow} + p_j m_j^{\downarrow} = \max_{\mathcal{M}} \left\{ \sum_j \mathbb{E}_{\mathbf{c}}[m_j^{\mathcal{M}}], \text{ subject to } \mathcal{M} \text{ being feasible and IR} \right\}.$$

**Remark 5** (Decreased data collection). *By construction of our mechanism, the contribution of a high-cost agent would be* $m_i^{\uparrow} \in [\bar{m}^*, \bar{m}^{\max}]$ *i.e. they contribute more than they would on their own, but lesser than the max possible under known costs. Further, our assumption that* $v(\cdot)$ *is concave means* $v'(\cdot)$ *is non-increasing. Hence,* (16) *implies that the data contributed by a low-cost agent is* $m_i^{\downarrow} \in [\bar{m}^{\max}, \underline{m}^{\max}]$. *However, if* $p_i \geq \frac{\underline{c}}{\underline{c} + \bar{c}}$, (15) *always implies that* $m_i^{\downarrow} = \underline{m}^{\max}$.

We extract lesser data than if we knew the agent's true cost i.e. $m_i^{\uparrow} \leq \bar{m}^{\max}$. However, they also receive a model which has worse value with $v(\bar{m}^*) + \bar{c}(m_i^{\uparrow} - \bar{m}^*) \leq v(\sum_j m_j)$ i.e. it is not trained on the combined data. This is because if we offered a full value model to a high-cost agent at $\bar{m}^{\max} \leq m_i^{\downarrow}$ contribution, the low-cost agent can claim they are actually high-cost and cheat our system. Instead, now the low-cost agent will contribute $m_i^{\downarrow} \geq \bar{m}^{\max}$ and will receive a model trained on the combined data with value $v(\sum_j m_j)$.

**Theorem VIII** (Information rent). *Consider our optimal mechanism* (15) *with equilibrium contributions* $m_i^{eq} = m_i^{\uparrow}$ *for a high-cost agent and* $m_i^{eq} = m_i^{\downarrow}$ *for the low-cost agent. Further, let* $\bar{m}^*$ *and* $\underline{m}^*$ *be the equilibrium individual contributions. Then, the utility of the high-cost agent remains unchanged with* $v(\bar{m}^*) + \bar{c}(m_i^{\uparrow} - \bar{m}^*) - \bar{c}m_i^{\uparrow} = v(\bar{m}^*) - \bar{c}\bar{m}^*$. *The utility of a low-cost agent, however, improves by* $\left( \underline{c}(\underline{m}^{\max} - m_i^{\downarrow}) - v(\underline{m}^{\max} + \Delta m_{-i}) + v(m_i^{\downarrow} + \Delta m_{-i}) \right) \geq 0$.

Because a low-cost agent can always lie and pretend to be high cost, they hold some power over the server when $m_i^{\downarrow} < \underline{m}^{\max}$. This is reflected in the extra utility they manage to extract and is called information rent. The utility of the high-cost agent remains unchanged since they hold no such power.

 # E   Proofs from Section 2 (Optimal Individual Contributions)

678 **Theorem I** (Optimal individual generation). *Consider an individual agent $i$ with marginal cost per*
679 *data point $c_i$ and value function $v$ satisfying* (1) *working on their own. Then, the optimal amount of*
680 *data $m_i^*$ is:*

$$m_i^* = \begin{cases} 0 & \text{if } \max_{m_i \geq 0} u_i(m_i) \leq 0; \\ \alpha_i^*, \text{ such that } b'(\alpha_i^*) = c_i & \text{otherwise.} \end{cases} \tag{4}$$

681 *Further, for agents $i, j$ with costs $c_i \leq c_j$, their utility satisfies $u_i(m_i^*) \geq u_j(m_j^*)$ and $m_i^* \geq m_j^*$.*

682 *Proof.* Recall that the utility function (see Eq. 3) of a single agent is:

$$u_i(m_i) = v(m_i) - c_i m_i.$$

683 Thus we have,

$$u_i'(m_i) = v'(m_i) - c_i.$$

684 Denote $\arg\max_m v(m) = 0$ as $m^0$. By definition, $\forall m_i > m^0$, $v(m_i) = b(m_i) > 0$. Given that $b(\cdot)$
685 is concave, $b'(m_i), m_i \geq m^0$ (or $u_i'(m_i)$) is maximized when $m_i = m^0$.

686 **Case 1 (high-cost agent):** $u_i'(m^0) \leq 0$. Then, for $\forall m_i \geq m^0$, $u_i'(m_i) \leq u_i'(m^0) \leq 0$. On the
687 other hand, $\forall 0 \leq m_i \leq m^0$, $u_i'(m_i) = -c_i \leq 0$. Thus $u_i(m_i)$ is non-increasing, and $m_i^* = 0$. The
688 utility function of an agent in this case is illustrated in Figure 1 (d).

689 **Case 2 (mid-cost agent):** $u_i'(m^0) > 0$ **and** $\max_{m_i} u_i(m_i) \leq 0$. When $u_i'(m^0) > 0$, that implies
690 that at $m^0$, $b'(m^0) > c_i$. Moreover, for $m_i \geq m^0$, we have that

$$u_i'(m_i) = b'(m_i) - c_i < b'(m^0) - c_i.$$

691 Therefore, since $b(m_i)$ is concave, it is possible that $u_i(m_i)$ increases first after $m^0$. However, as
692 long as $\max_{m_i} u_i(m_i) \leq 0$, we still have that $m_i^* = 0$. The utility function of an agent in this case is
693 illustrated in Figure 1 (c).

694 **Case 3 (low-cost agent):** $u_i'(m^0) > 0$ **and** $\max_{m_i} u_i(m_i) > 0$. Recall that for a mid-cost agent, it
695 is possible that $u_i(m_i)$ increases first after $m^0$. Moreover, given that $v(m_i) \leq 1$, as $m_i \to \infty$,

$$u_i'(m_i) = b'(m_i) - c_i \leq 0.$$

696 Therefore, there exists $\alpha_i^* > m^0 > 0$ such that $b'(\alpha_i^*) = c_i$. The utility function of an agent in this
697 case is illustrated in Figure 1 (b).

698 Combining the three cases above completes the first part of the proof.

699 Next, consider two agents with costs $c_i \leq c_j$. Note that for any fixed $m$, $u_i(m) \geq u_j(m)$. Hence,
700 the inequality also holds after minimizing both sides. Finally, note that if $j$ is not a low-cost agent,
701 it is clear that $m_i^* \geq m_j^* = 0$. If both $i$ and $j$ are low-cost agents, note that $m_i^* = b'^{-1}(c_i)$ and
702 $m_j^* = b'^{-1}(c_j)$. Since $b(\cdot)$ is concave and positive, $b'$ (and hence $b'^{-1}$) is non-increasing. This
703 implies that $m_i^* \geq m_j^*$ finishing the theorem. $\square$

704 # F   Proofs from Section 3 (Modeling Multiple Agents and Catastrophic
705 Free-riding)

706 **Theorem II** (Existence of pure equilibrium). *Consider a feasible mechanism $\mathcal{M}$ which can be*
707 *expressed as:*

$$[\mathcal{M}(m_i; \mathbf{m}_{-i})]_i = \max(0, \nu_i(m_i; \mathbf{m}_{-i})),$$

708 *for a function $\nu_i(m_i; \mathbf{m}_{-i})$ which is* continuous *in $\mathbf{m}$ and* concave *in $m_i$. For any such $\mathcal{M}$, there*
709 *exists a pure Nash equilibrium in data contributions $\mathbf{m}^{eq}(\mathcal{M})$ which for any agent $i$ satisfies,*

$$[\mathcal{M}(\mathbf{m}^{eq}(\mathcal{M}))]_i - c_i m_i^{\mathcal{M}} \geq [\mathcal{M}(m_i, \mathbf{m}^{eq}(\mathcal{M})_{-i})]_i - c_i m_i, \text{ for all } m_i \geq 0. \tag{7}$$

*Proof.* For a set of contributions $\mathbf{m}$, define the following best response mapping:

$$[B(\mathbf{m})]_i := \arg\max_{\tilde{m}_i \geq 0} \{u_i(\tilde{m}_i, \mathbf{m}_{-i}) := [\mathcal{M}(\tilde{m}_i, \mathbf{m}_{-i})]_i - c_i\tilde{m}_i\}, \tag{18}$$

where recall $[\mathcal{M}(\tilde{m}_i, \mathbf{m}_{-i})]_i$ is the value returned by the mechanism upon agent $i$ submitting $\tilde{m}_i$ data points and the rest contributing $\mathbf{m}_{-i}$. Note that the mapping defined above is a multi-valued function i.e. $B : \mathbb{R}^n \to 2^{\mathbb{R}^n}$. This is because the $\arg\max$ defined above may potentially return multiple values. Nevertheless, suppose that there existed a fixed point to the mapping $B$ i.e. there existed $\tilde{\mathbf{m}}$ such that $\tilde{\mathbf{m}} \in B(\tilde{\mathbf{m}})$. Then, $\tilde{\mathbf{m}}$ is the required equilibrium contribution since by definition of the arg-max we have for any $m_i \geq 0$,

$$[\mathcal{M}(\tilde{m}_i, \tilde{\mathbf{m}}_{-i})]_i - c_i\tilde{m}_i \geq [\mathcal{M}(m_i, \tilde{\mathbf{m}}_{-i})]_i - c_i m_i.$$

So, we only have to prove that the mapping $B$ has a fixed point. Since the mechanism $\mathcal{M}$ is feasible, by Definition A and equation (1) we have

$$[\mathcal{M}(\mathbf{m})]_i \leq v(\textstyle\sum_j m_j) \leq \lim_{m\to\infty} v(m) \leq 1.$$

This implies that

$$0 \geq u_i(\tilde{\mathbf{m}}) \leq 1 - c_i\tilde{m}_i \Rightarrow \tilde{m}_i \leq 1/c_i.$$

Thus, we can restrict our search space to a *convex* and *compact* product set $\mathcal{C} := \prod_j[0, 1/c_i] \subset \mathbb{R}^n$ and our mapping is then over $B : \mathcal{C} \to 2^{\mathcal{C}}$. Next by assumption on the mechanism $\mathcal{M}$, our utility function can be written as

$$u_i(m_i, \mathbf{m}_{-i}) = \max(-c_i m_i, \nu(m_i, \mathbf{m}_{-i}) - c_i m_i),$$

where $\nu(m_i, \mathbf{m}_{-i}) - c_i m_i$ is concave in $m_i$. Unfortunately, $u_i$ may not be quasi-concave in $m_i$ because of the max. If it was quasi-concave, the mapping $B(\mathbf{m})$ would be continuous in $\mathbf{m}$ and applying Kakutani's theorem would yield the existence of the required fixed point (see Maskin [36] or Acemoglu and Ozdaglar [3, Lecture 11], for details).

**Lemma 6** (Kakutani's fixed point theorem). *Consider a multi-valued function $F : \mathcal{C} \to 2^{\mathcal{C}}$ over convex and compact domain $\mathcal{C}$ for which the output set $F(\mathbf{m})$ i) is convex and closed for any fixed $\mathbf{m}$, and ii) changes continuously as we change $\mathbf{m}$. For any such $F$, there exists a fixed point $\mathbf{m}$ such that $\mathbf{m} \in F(\mathbf{m})$.*

However, our utility function is not quasi-concave and the mapping $B$ may be discontinuous. While there have been recent extensions of Kakutani's fixed point theorem to half-continuous functions (e.g. Bich [6, Theorem 3.2]), the mapping $B$ does not satisfy this either. We next study the exact nature of discontinuity.

**Lemma 7.** *Consider the best-response mapping $B$ in (18) over convex and compact domain $\mathcal{C}$. For any $\mathbf{m}$, either the mapping $[B(\mathbf{m})]_i$ is convex, closed, and continuous in $\mathbf{m}$, or $0 \in [B(\mathbf{m})]_i$.*

*Proof.* Figure 9 looks at the best response mapping $B_i$ depending on the utility curve $u_i(\cdot, \mathbf{m}_{-i})$. Even if the utility itself is smoothly varying with the parameters $\mathbf{m}$, the best response may be discontinuous. In Fig. 9, for a small change in the utility curve between $u2(m_i)$ to $u3(m_i)$, the best response drastically changes from $\tilde{m}_i = 0$ (A) to $\tilde{m}_i > 125$ (B). However, this is the only source of discontinuity.

Recall that our utility function $u_i$ is a max of a decreasing linear function and a concave function. Thus it has at most two local maxima: either 0, or the maxima of the concave function $f(m_i) = \nu_i(m_i; \mathbf{m}_{-i}) - c_i m_i$. The set of maxima of a continuous concave function is continuous, closed and convex. Hence, either 0 is part of the best response, or $[B(\mathbf{m})]_i$ is continuous, closed and convex. $\square$

Armed with Kakutani's fixed point theorem Lemma 6 and a description of the discontinuities in the best response mapping Lemma 7, we can continue with the proof of existence of a fixed point for $B$. Given any index set $\mathcal{I} \subseteq [n]$, we can define the following sub-domain $\mathcal{C}_{\mathcal{I}} := \prod_{i \in \mathcal{I}}[0, 1/c_i]$. Given any vector $\mathbf{p} \in \mathcal{C}_{\mathcal{I}}$, we can construct its extension $m(\mathbf{p}; \mathcal{I}) \in \mathcal{C}$ as

$$[m(\mathbf{p}; \mathcal{I})]_i := \mathbf{p}_i \text{ if } i \in \mathcal{I}, \text{ and } 0 \text{ otherwise.}$$

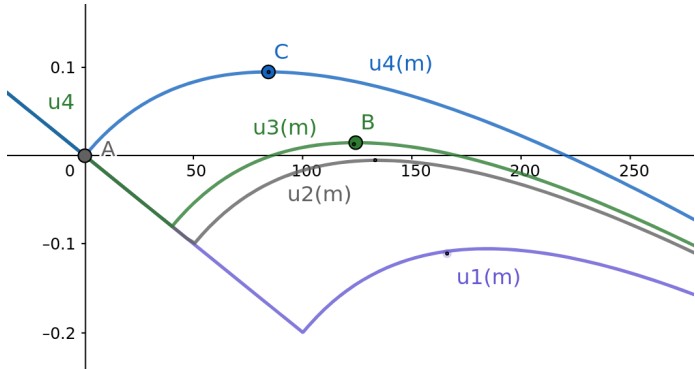

Figure 9: Utility curves $u(m_i)$ of some agent $i$, and the corresponding discontinuous best responses ($\tilde{m}_i \geq 0$ which maximizes $u(m_i)$). For both $u1$ and $u2$, the best response of the agent is $\tilde{m}_i = 0$ (point A), and points B and C are the best responses for $u3$ and $u4$. A small change in the utility curves (from $u2$ to $u3$) can result in a large change in the best response (from A to B).

We will omit the $\mathcal{I}$ dependence and use $m(\mathbf{p})$ when clear from context. Given this mapping between sub-domain $\mathcal{C}_\mathcal{I}$ and the full domain $\mathcal{C}$, we can define a mapping:

$$B_\mathcal{I}(\mathbf{p}) : \mathcal{C}_\mathcal{I} \to 2^{\mathcal{C}_\mathcal{I}} := ([B(m(\mathbf{p}))]_i \text{ for } i \in \mathcal{I})$$

Finally, for any $\mathbf{m} \in \mathcal{C}$, define the set of indices $\mathcal{I}(\mathbf{m}) \subseteq [n]$ as

$$\mathcal{I}(\mathbf{m}) := \{i \text{ for which } 0 \notin [B(\mathbf{m})]_i .\}$$

Let us start from $\mathbf{m} = \mathbf{0}$. If $\mathcal{I}(\mathbf{0}) = \emptyset$, we are done since this implies $\mathbf{0} \in B(\mathbf{0})$. Otherwise, Lemma 7 states that the mapping $B_{\mathcal{I}(\mathbf{0})}(\mathbf{p})$ over the compact convex domain $\mathcal{C}_{\mathcal{I}(\mathbf{0})}$ is convex, compact and continuous. Hence, by Lemma 6, it has a fixed point such that $\mathbf{p}^1 \in B_\mathcal{I}(\mathbf{p}^1)$. We can inductively continue applying the same argument. If $m(\mathbf{p}^1)$ is a fixed point of the full mapping $B$ with $m(\mathbf{p}^1) \in B(m(\mathbf{p}^1))$, we are done. Otherwise, $\mathcal{I}(m(\mathbf{p}^1)) \supset \mathcal{I}(\mathbf{0})$ and we can continue repeating the same argument inductively. Since the size of $\mathcal{I}$ is at most $n$, the recursion will stop and yield a fixed point $\tilde{\mathbf{m}} \in \mathcal{C}$ such that $\tilde{\mathbf{m}} \in B(\tilde{\mathbf{m}})$. As we initially proved, this fixed point $\tilde{\mathbf{m}}$ to the best response dynamics is also the equilibrium of our mechanism.

$\square$

**Theorem III** (Catastrophic free-riding). *Consider $n$ agents with costs $\{c_i\}$ with a unique least cost agent $c_{\min} = \min_i c_i$. Let $\{m_i^*\}$ be the equilibrium contributions of agents when alone. The standard federated learning mechanism corresponding to $[\mathcal{M}(\mathbf{m})]_i = v(\sum_j m_j)$ for all clients $i$ is feasible and IR, and has an unique equilibrium. At this equilibrium, only the lowest cost agent contributes:*

$$m_i^{eq} = \begin{cases} m_i^* & \text{if } c_i = c_{\min} \\ 0 & \text{otherwise.} \end{cases} \tag{9}$$

*Proof.* Let $\tilde{i}$ be the agent with the least cost. By Theorem I, it follows that $m_j^* < m_{\tilde{i}}^*$ for all agents $j \in [n]$.

First, suppose that $m_{\tilde{i}}^* > 0$. In this setting, all other agents agent $j$ will have access to data contributed by $\tilde{i}$ which is $m_{\tilde{i}}^*$. Now given access to this, the marginal gain in value for an additional data-point for any agent $j$ is less than their cost i.e. $b'(m_{\tilde{i}}^*) = c_{\tilde{i}} < c_j$. Hence, it is optimal for agent $j$ to just use $m_{\tilde{i}}^*$ data-points, and not generate any additional data-points. Thus, the equilibrium is all other agents contribute no data, and agent $\tilde{i}$ computes $m_{\tilde{i}}^*$ datapoints as if on its own.

Next consider the case where $m_{\tilde{i}}^* = 0$ and hence all $m_i^* = 0$. Suppose all the other agents in total contribute $\Delta m$ datapoints, which is given to all agents unconditionally. With this extra free data, it is possible that there exists some agent for whom $m_{\tilde{i}}^* > 0$. However, the incentives of the agents remain identical and so the agent with the least cost collects the most data. Given access to this $\Delta m + m_{\tilde{i}}^*$ amount of data, agent $j$ with higher cost $c_j \geq c_{\tilde{i}}$ has no incentive to collect any additional

778 data. Hence, only agent $\tilde{i}$ would collect any data and so $\Delta m = 0$. However, if $\Delta m = 0$, agent $\tilde{i}$ also
779 has no incentive to collect any data. This implies that all agents contributing $0$ datapoints is the only
780 Nash equilibrium possible. □

# G Proofs from Section 4 (value Shaping under Known Costs)

782 **Theorem IV** (Data maximization with known costs). *The mechanism $\mathcal{M}$ defined by* (11) *is data-*
783 *maximizing for $\varepsilon \to 0^+$. At equilibrium, a rational agent $i$ will contribute $m_i^{\max}$ data points where*
784 $m_i^{\max} \geq m_i^*$, *yielding a total of $\sum_j m_j^{\max}$ data points.*

785 **Proof.** We will do the proof in two steps. Consider the best response $B_{\mathcal{M}}$ of the agents to a
786 mechanism $\mathcal{M}$ similar to the definition in the proof of Theorem II:

$$[B_{\mathcal{M}}(\mathbf{m})]_i := \arg\max_{m_i \geq 0}[\mathcal{M}(m_i, \mathbf{m}_{-i})]_i - c_i m_i \,.$$

787 We will first prove that given a fixed contribution $\mathbf{m}$ from other users, the best response for our
788 mechanism $m_i^{\max}$ is higher than that of any other feasible and IR mechanism. We will then show that
789 this necessarily implies that the equilibrium contribution of the agent is also data maximizing.

790 **Lemma 8.** *For a given data contribution $\mathbf{m}$ and any feasible and IR mechanism $\tilde{\mathcal{M}}$, define best*
791 *responses $B_{\mathcal{M}}(\mathbf{m})$ and $B_{\tilde{\mathcal{M}}}(\mathbf{m})$ for our mechanism $\mathcal{M}$ (defined in* (11)) *and the other mechanism*
792 $\tilde{\mathcal{M}}$. *Then, for any agent $i$ and contribution $\mathbf{m}$,*

$$[B_{\mathcal{M}}(\mathbf{m})]_i \geq [B_{\tilde{\mathcal{M}}}(\mathbf{m})]_i \,.$$

793 *Further, the best response $[B_{\mathcal{M}}(\mathbf{m})]_i$ is non-decreasing in the net contribution from other agents*
794 $(\sum_{j \neq i} \mathbf{m}_j)$.

795 For now, we will assume that the above lemma and continue with our proof. As shown in the proof of
796 Theorem II, the equilibrium of all feasible mechanisms (if they exist) lie in the range $\mathcal{C} := \prod_i [0, 1/c_i]$.
797 Suppose that $\tilde{\mathbf{m}} \in \mathcal{M}$ is the equilibrium of mechanism $\tilde{\mathcal{M}} \in \mathcal{M}$. Note that $\tilde{\mathbf{m}}$ is also the fixed point
798 of the best response with $\tilde{\mathbf{m}} \in B_{\tilde{\mathcal{M}}}(\tilde{\mathbf{m}})$. Now, define the following subspace

$$\mathcal{C}_{\geq \tilde{\mathbf{m}}} := \prod_j [\tilde{\mathbf{m}}_j, 1/c_j] \,.$$

799 The set $\mathcal{C}_{\geq \tilde{\mathbf{m}}}$ is compact and convex. Thus, we can apply Theorem II to our optimal mechanism $\mathcal{M}$
800 to prove that there exists an equilibrium point $\mathbf{m} \in \mathcal{C}_{\geq \tilde{\mathbf{m}}}$ such that

$$[\mathbf{m}]_i \in \arg\max_{m_i \geq \tilde{m}_i}[\mathcal{M}(m_i, \mathbf{m}_{-i})]_i - c_i m_i \,.$$

801 We will next show that the above point $\mathbf{m}$ is in fact a fixed of $B_{\mathcal{M}}(\mathbf{m})$ and satisfies:

$$[\mathbf{m}]_i \in \arg\max_{m_i \geq 0}[\mathcal{M}(m_i, \mathbf{m}_{-i})]_i - c_i m_i \,.$$

802 Note that the only difference between the two claims is that in the latter the $\arg\max$ is taken over
803 $\geq 0$ where as it was more constrained in the former. For the sake of contradiction, suppose this is
804 not true i.e. there exists an agent $i$ such that $\mathbf{m}_i \notin [B_{\mathcal{M}}(\mathbf{m})]_i$ and $[B_{\mathcal{M}}(\mathbf{m})]_i < \tilde{\mathbf{m}}_i$. However, this
805 leads to a contradiction:

$$\sum_{j \neq i} \mathbf{m}_j \geq \sum_{j \neq i} \tilde{\mathbf{m}}_j$$
$$\Rightarrow [B_{\mathcal{M}}(\mathbf{m})]_i \geq [B_{\mathcal{M}}(\tilde{\mathbf{m}})]_i \geq [B_{\tilde{\mathcal{M}}}(\tilde{\mathbf{m}})]_i = \tilde{\mathbf{m}}_i \,.$$

806 The first inequality is because $\mathbf{m} \in \mathcal{C}_{\geq \tilde{\mathbf{m}}}$. The first inequality in the second step follows from the
807 latter part of Lemma 8 while the next inequality is from the first part. Finally, the last equality
808 follows because $\tilde{\mathbf{m}}$ is a fixed point of $B_{\tilde{\mathcal{M}}}$. Hence, we have proven that there exists a fixed point
809 $\mathbf{m} \in B_{\mathcal{M}}(\mathbf{m})$ such that $\mathbf{m} \in \mathcal{C}_{\geq \tilde{\mathbf{m}}}$ i.e. the equilibrium contribution of every agent under $\mathcal{M}$ is at
810 least as much as $\tilde{\mathcal{M}}$. □

811 **Proof of Lemma 8.** Recall the optimal mechanism defined in (11) restated below:

$$[\mathcal{M}(\mathbf{m})]_i = \begin{cases} v(m_i) & \text{for } m_i \leq m_i^* \\ v(m_i^*) + (c_i + \varepsilon)(m_i - m_i^*) & \text{for } m_i \in [m_i^*, m_i^{\max}] \\ v(\sum_j m_j) & \text{for } m_i \geq m_i^{\max} . \end{cases} \quad (19)$$

812 For now, suppose that $m_i^* > 0$. Recall, from [case 3, Theorem I], that this implies $v'(m_i^*) =$
813 $b'(m_i^*) = c_i$.

814 First we show that $m_i^{\max}$ is the unique equilibrium contribution for an agent $i$. The slope of the utility
815 of agent $i$ is

$$u_i'(m_i; \mathcal{M}) = \frac{\partial [\mathcal{M}(\mathbf{m})]_i}{\partial m_i} - c_i .$$

816 By construction, this slope is $u_i'(m_i; \mathcal{M}) > 0$ for any $m_i < m_i^{\max}$. Suppose the contribution of
817 all other agents is fixed to $\Delta m_{-i} = \sum_{j \neq i} m_j$. The slope of the utility at $m_i^{\max}$ is $u_i'(m_i^{\max}; \mathcal{M}) =$
818 $v'(m_i^{\max} + \Delta m_{-i}) - c_i$. Again, by construction, $m_i^{\max} + \Delta m_{-i} \geq m_i^*$. Since $b$ is concave and $b'$ is
819 non-increasing,

$$u_i'(m_i^{\max}; \mathcal{M}) = v'(m_i^{\max} + \Delta m_{-i}) - c_i = b'(m_i^{\max} + \Delta m_{-i}) - c_i \leq b'(m_i^*) - c_i = 0 .$$

820 Thus, $m_i^{\max}$ is the unique equilibrium contribution of agent $i$. Next, we have to demonstrate the
821 data-maximizing property. For the sake of contradiction, suppose there existed some other mechanism
822 $\tilde{\mathcal{M}}$ such that

$$\arg\max_{m_i} [\tilde{\mathcal{M}}(\mathbf{m})]_i - c_i m_i =: \tilde{m}_i > m_i^{\max} .$$

823 This implies that $u_i'(m_i; \tilde{\mathcal{M}}) > 0$ for any $m_i \leq \tilde{m}_i$, i.e. $\frac{\partial [\tilde{\mathcal{M}}(\mathbf{m})]_i}{\partial m_i} > c_i$. In particular, this implies
824 that

$$\frac{\partial [\tilde{\mathcal{M}}(\mathbf{m})]_i}{\partial m_i} > \frac{\partial [\mathcal{M}(\mathbf{m})]_i}{\partial m_i} \text{ for all } m_i \in [m_i^*, m_i^{\max}] .$$

825 Further, $\tilde{\mathcal{M}}$ satisfies individual rationality and so at $m_i = m_i^*$ we have

$$[\tilde{\mathcal{M}}(m_i^*, \mathbf{m}_{-i})]_i \geq v(m_i^*) = [\mathcal{M}(m_i^*, \mathbf{m}_{-i})]_i .$$

826 Together, these two conditions imply that for all $m_i \in [m_i^*, m_i^{\max}]$, we have $[\tilde{\mathcal{M}}(\mathbf{m})]_i > [\mathcal{M}(\mathbf{m})]_i$.
827 In particular at $m_i = m_i^{\max}$, we have

$$[\tilde{\mathcal{M}}(m_i^{\max}, \mathbf{m}_{-i})]_i > v(\sum_j m_j) .$$

828 This gives us a contradiction since it violates feasibility. Thus, $m_i^{\max}$ is the maximum data which can
829 be extracted from agent $i$.

830 The proofs for the low and medium cost agents are similar, while noting that $m_i^* = 0$. This finishes
831 the proof of the first part. The second part of the lemma follows directly from the definition of $\mathcal{M}$ and
832 the fact that the value function $v(m_i + \sum_{j \neq i} m_j)$ is non-decreasing in the contributions $\sum_{j \neq i} m_j$.
833 $\square$

834 **Theorem V** (Incentive compatibility). *Under given costs* $\mathbf{c}$, *consider our optimal mechanism* (11)
835 *with equilibrium contributions* $\mathbf{m}^{\max}$, *and agents working individually with equilibrium contributions*
836 *of* $\mathbf{m}^*$. *The utility of the every agent* $i$ *remains unchanged:*

$$v(\sum_j m_j^{\max}) - c_i m_i^{\max} = v(m_i^*) - c_i m_i^* .$$

837 *Proof.* This statement is true by construction of our mechanism. When $\varepsilon \to 0$, the slope of the utility
838 becomes

$$u_i'(m_i; \tilde{\mathcal{M}}) = \frac{\partial [\mathcal{M}(\mathbf{m})]_i}{\partial m_i} - c_i = c_i + \varepsilon - c_i = 0 \text{ for all } m_i \in [m_i^*, m_i^{\max}] .$$

839 Further, note that at $m_i = m_i^*$, we have $[\mathcal{M}(m_i^*, \mathbf{m}_{-i})]_i = v(m_i^*)$. Thus, for all $m_i \in [m_i^*, m_i^{\max}]$,
840 the utility of agent $i$ with our mechanism $\mathcal{M}$ remains constant and equal to the optimal individual
841 utility $u_i(m_i^*)$. $\square$

## H   Proofs from Appendix D (Data Maximization with Unverifiable Costs)

**Theorem VII** (Expected data maximization). *Mechanism* (15) *is feasible, satisfies IR, and has a unique Nash equilibrium:* $m_i^{eq} = m_i^{\uparrow}$ *if* $c_i = \bar{c}$ *and otherwise* $m_i^{eq} = m_i^{\downarrow}$. *Further, for* $\varepsilon \to 0^+$, *the mechanism* (15) *maximizes the expected (over the sampling of the true costs) amount of data collected with*

$$\sum_j (1 - p_j) m_j^{\uparrow} + p_j m_j^{\downarrow} = \max_{\mathcal{M}} \Big\{ \sum_j \mathbb{E}_{\mathbf{c}}[m_j^{\mathcal{M}}] , \text{ subject to } \mathcal{M} \text{ being feasible and IR} \Big\} .$$

*Proof.* Recall that we had defined the mechanism (15) as

$$[\mathcal{M}(\mathbf{m})]_i = \begin{cases} v(m_i) & \text{for } m_i \leq \bar{m}^* \\ v(\bar{m}^*) + (\bar{c} + \varepsilon)(m_i - \bar{m}^*) & \text{for } m_i \in [\bar{m}^*, m_i^{\uparrow}] \\ v(m_i^{\downarrow} + \sum_{j \neq i} m_j) - (\underline{c} + \varepsilon)(m_i^{\downarrow} - m_i) & \text{for } m_i \in [m_i^{\uparrow}, m_i^{\downarrow}] \\ v(\sum_j m_j) & \text{for } m_i \geq m_i^{\downarrow} . \end{cases} \tag{20}$$

First, we have to show that $m_i^{\uparrow}$ and $m_i^{\downarrow}$ are equilibrium for the high and low cost players $\bar{c}$ and $\underline{c}$ respectively. For the sake of simplicity, we first assume that $\bar{m}^* > 0$ and $\underline{m}^* > 0$. The proofs directly extend to the other cases. Now, note that $v'(\bar{m}^*) = b'(\bar{m}^*) = \bar{c}$. Thus, by constructions, we have that for a high cost agent,

$$u_i'(m_i; \mathcal{M}) = \frac{\partial [\mathcal{M}(\mathbf{m})]_i}{\partial m_i} - \bar{c} > 0 \text{ for all } m_i \leq m_i^{\uparrow} .$$

where as for $m_i > m_i^{\uparrow}$, the slope $u_i'(m_i; \mathcal{M}) = \underline{c} + \varepsilon - \bar{c} < 0$. Assuming $\varepsilon$ is small enough, a high cost agent obtains optimal utility at $m_i^{\uparrow}$. Similarly, for the low cost agent, $u_i'(m_i; \mathcal{M}) > 0$ for all $m_i < m_i^{\downarrow}$ and is negative after (similar to Theorem IV). Thus, the optimum contribution of the low cost player is $m_i^{\downarrow}$.

Next, recall that we had defined in (16) that $m_i^{\downarrow}$ satisfies

$$m_i^{\downarrow} = \min\Big( \max\Big( b'^{-1}\Big( \underline{c} - \tfrac{p_i}{1-p_i} \bar{c} \Big) - \Delta m_{\text{-}i} , \bar{m}^{\max} \Big), \underline{m}^{\max} \Big) . \tag{21}$$

We will show that $m_i^{\downarrow}$ defined this way maximizes the expected data for agent $i$:

$$\max_{m_i^{\downarrow}} \Big\{ (1 - p_i) m_i^{\uparrow} + p_i m_i^{\downarrow} \Big\} \text{ subject to } m_i^{\uparrow}, m_i^{\downarrow} \text{ are feasible for } \mathcal{M} . \tag{22}$$

This involves some variational calculus (see Fig. 10). As shown in Fig. 10, reducing the value of $m_i^{\downarrow}$ results in an increase in $m_i^{\uparrow}$. Suppose we push the blue bar vertically by a small value $dx$. Because the slope of AC is $\bar{c}$, this results in increase of $\frac{dx}{\bar{c}}$ in $m_i^{\uparrow}$. Correspondingly, we can show that the decrease in $m_i^{\downarrow}$ will be $\frac{dx}{\underline{c} - v'(m_i^{\downarrow} + \Delta m_{\text{-}i})}$. Putting these together, the net expected change in data contribution is

$$(1 - p_i) \frac{dx}{\bar{c}} - \frac{dx}{\underline{c} - b'(m_i^{\downarrow} + \Delta m_{\text{-}i})} .$$

The local unconstrained maxima can then be derived by setting the above to 0 i.e when

$$b'(m_i^{\downarrow} + \Delta m_{\text{-}i}) = \underline{c} - \tfrac{p_i}{1-p_i} \bar{c} .$$

Of course, we have to respect the constraints that $m_i^{\downarrow} \in [\bar{m}^{\max}, \underline{m}^{\max}]$ giving us our final result. Thus, the value of $m_i^{\downarrow}$ as chosen by (22) is optimal for these class of mechanisms.

Now, we have to show that any data-maximizing mechanism corresponds to $\mathcal{M}$ with some choice of $m_i^{\downarrow}$. Consider a mechanism $\tilde{\mathcal{M}}$ whose equilibrium contributions are $(\tilde{m}, \underline{m})$ for a high and low-cost agent respectively (see points I and J in Fig. 10). Now, from the optimality of $\underline{m}^{\max}$, we know that $\underline{m} \leq \underline{m}^{\max}$. Let us connect $\bar{m}^*$ (point A) to $\tilde{m}$ (point I) and then to $\underline{m}$ (point J). Recall that we assumed that $\tilde{\mathcal{M}}$ is different from $\mathcal{M}$ in (15). This means that the slope AI $\neq \bar{c}$ or IJ $\neq \underline{c}$. Consider

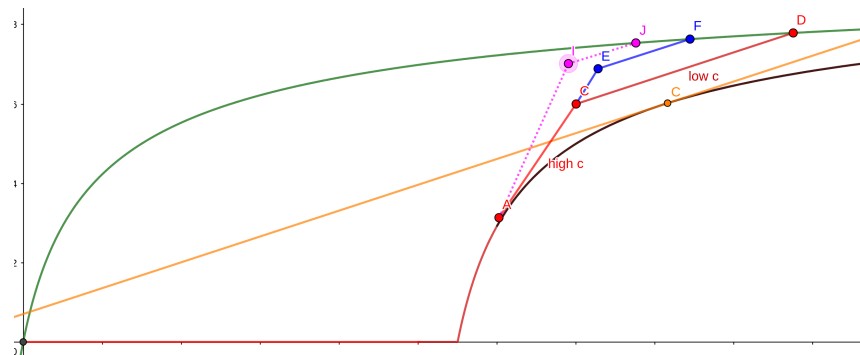

Figure 10: Value shaping mechanism under unknown costs. *(red curve)*: model value returned to agent $i$ by the mechanism; *(grey curve)*: model value for agent $i$ without participation; *(green curve)*: model value if agent $i$ receives all the data from the other agents. Point C and D (in red) and points E and F (in blue) represent two different choices for ($m^\uparrow$ and $m^\downarrow$) respectively. If we choose a smaller value of $m^\downarrow$ (shown in blue by point F), we would see an increase in $m^\uparrow$ to point E. Thus, the optimum value balances these two depending on the probability $p_i$. Finally, points I and J (in magenta) represent potential other mechanisms $\tilde{\mathcal{M}}$.

870    the latter. Combined with I and J corresponding to equilibria, we have slope of IJ $> \underline{c}$. This implies
871    that starting from point I, we could have instead drawn a line segment of slope $\underline{c}$ and increased the
872    data contribution by the low cost agent, while keeping the contribution of the high-cost agent fixed.
873    Similarly, we can show that the optimal slope for AI is $\bar{c}$. Together, this implies that any optimal
874    mechanism $\mathcal{M}$ must be of the form (15), finishing our proof.    $\square$

875    **Theorem VIII** (Information rent). *Consider our optimal mechanism* (15) *with equilibrium contri-*
876    *butions $m_i^{eq} = m_i^\uparrow$ for a high-cost agent and $m_i^{eq} = m_i^\downarrow$ for the low-cost agent. Further, let $\bar{m}^*$*
877    *and $\underline{m}^*$ be the equilibrium individual contributions. Then, the utility of the high-cost agent remains*
878    *unchanged with $v(\bar{m}^*) + \bar{c}(m_i^\uparrow - \bar{m}^*) - \bar{c}m_i^\uparrow = v(\bar{m}^*) - \bar{c}\bar{m}^*$. The utility of a low-cost agent,*
879    *however, improves by $\left( \underline{c}(\underline{m}^{\max} - m_i^\downarrow) - v(\underline{m}^{\max} + \Delta m_{-i}) + v(m_i^\downarrow + \Delta m_{-i}) \right) \geq 0$.*

880    *Proof.* For a high cost player, the statement easily follows since $\frac{\partial [\mathcal{M}(\mathbf{m})]_i}{\partial m_i} - \bar{c} = 0$ for all $m_i \in [\bar{c}^*, c_i^\uparrow]$.
881    Thus, a high cost player's utility remains constant during this period and is equal to utility at $m_i = \bar{m}^*$
882    which is $v(\bar{m}^*) - \bar{c}\bar{m}^*$.

883    For a low cost agent, $\frac{\partial [\mathcal{M}(\mathbf{m})]_i}{\partial m_i} - \bar{c} = 0$ for all $m_i \in [m_i^\uparrow, m_i^\downarrow]$, and hence their utility is constant in
884    this region. In particular, the difference in utility with mechanism $\mathcal{M}$ and alone is

$$v(m_i^\downarrow + \Delta m_{-i}) - \underline{c}m_i^\downarrow - v(\underline{m}^{\max} + \Delta m_{-i}) + \underline{c}\underline{m}^{\max}.$$

885    The above quantity is always non-negative since $v'(m_i + \Delta m_{-i}) \leq \underline{c}$ for all $m_i \in [m_i^\downarrow, \underline{m}^{\max}]$.    $\square$

