# OpenReview forum: "Mechanisms that Incentivize Data Sharing in Federated Learning"
_NeurIPS.cc/2022/Workshop/Federated_Learning — FL-NeurIPS 2022 Oral_

### Official Review · Reviewer_3mFC · 2022-10-14
**Sound theoretical analysis but insufficient explaination and no experiments**

This paper analyzes the free-riding phenomenon in federated learning when rational clients may not be tempted to engage in federated learning. Using contract theory, the authors introduce accuracy shaping-based mechanisms to maximize the amount of data generated by each agent. There are too many long sentences making the paper hard to read. There are the following comments and questions:
1. The heterogeneity of the client's data in federated learning is ignored in the analysis by considering the same value (contribution) of each data sample.
2. The formulation throughout the paper is inconsistent. For example, since different clients have different values and data numbers, equation (3) should use $v_i$ and $m_i$.
3. In (4), what does $b'$ mean? Moreover, Theorem 1 should be explained more. It is confusing why $m*_I$ corresponds to when the value equals the cost.
4. Experiment is needed for the theoretical analysis.

---

### Official Review · Reviewer_XqBo · 2022-10-14
**Interesting attempt on a formal framework for incentivizing data sharing**

The paper studies way to incentivize clients in federated learning, so that they will contribute a maximal amount of data for training.

Pros:

- The paper studies an important topic: ways of rewarding clients in FL, so that a large amount of data can be collected. This ensures that all participants will learn a good model and that none of them will free-ride the collaborative learning process.
- The paper is very well-written, well-argumented and positioned clearly to prior work.
- The definitions related to the data distribution mechanism are fairly intuitive and the results are natural given the studied framework.
- Overall, the paper looks technically sound (see remarks below)

Weaknesses:

- For a person coming from statistics and machine learning, reading Section 2.1 was extremely confusing to me. When talking about Federated learning, one expects to see rewards written in terms of expected losses with respect to the target distribution. Once going over the whole of Section 2, the definitions make sense. However, given that the notation is fairly non-standard compared to the standard FL literature, it will be nice if the authors can expand on Section 2.1 and explain at the beginning how their setup fits in the standard Fl one.
- Related to this, I am a bit cautious of the used definition of a(m) as an upper confidence bound of the expect loss of the classifier. To me it seems more natural/desirable to consider a framework when one looks at the expected utilities instead, where the utility depends on the actual (test-time) loss of the resulting model and the randomness is with respect to the sampling of the train data. Could the authors comment on how this setup would differ from the one in the paper and if they expect this to lead a substantially harder analysis?
- For readability, it would be nice to define Nash equilibrium somewhere and explicitly mention that this is natural notion to study, given that the players are assumed to be rational.

---

### Official Review · Reviewer_7457 · 2022-10-19
**practical problem with a innovative solution**

A contract theory based framework is designed to associating the said cost and data generation.  The result of catastrophic free-riding is shown as giving unconditional benefit of combined data to all agents. Practical solution has been proposed to address the problem.

---

### Decision · Program_Chairs · 2022-10-20

Accept (Oral)